# Enhancing Bone Infection Diagnosis with Raman Handheld Spectroscopy: Pathogen Discrimination and Diagnostic Potential

**DOI:** 10.3390/ijms25010541

**Published:** 2023-12-30

**Authors:** Richard Andreas Lindtner, Alexander Wurm, Elena Pirchner, David Putzer, Rohit Arora, Débora Cristina Coraça-Huber, Michael Schirmer, Jovan Badzoka, Christoph Kappacher, Christian Wolfgang Huck, Johannes Dominikus Pallua

**Affiliations:** 1Department of Orthopaedics and Traumatology, Medical University of Innsbruck, Anichstraße 35, 6020 Innsbruck, Austria; richard.lindtner@i-med.ac.at (R.A.L.); alexanderwurm@a1.net (A.W.); elena.pirchner@student.i-med.ac.at (E.P.); david.putzer@i-med.ac.at (D.P.); rohit.arora@tirol-kliniken.at (R.A.); debora.coraca-huber@i-med.ac.at (D.C.C.-H.); 2Praxis Dr. Med. Univ. Alexander Wurm FA für Orthopädie und Traumatologie, Koflerweg 7, 6275 Stumm, Austria; 3Department of Internal Medicine, Clinic II, Medical University of Innsbruck, Anichstraße 35, 6020 Innsbruck, Austria; schirmer.michael@icloud.com; 4Institute of Analytical Chemistry and Radiochemistry, University of Innsbruck, Innrain 80-82, 6020 Innsbruck, Austria; jovan.badzoka@uibk.ac.at (J.B.); christoph.kappacher@uibk.ac.at (C.K.); christian.w.huck@uibk.ac.at (C.W.H.)

**Keywords:** bone quality, handheld Raman spectrometer, Raman spectroscopy, principal component analyses, *Staphylococcus aureus*, *Staphylococcus epidermidis*, bone infection, bone graft

## Abstract

Osteomyelitis is a bone disease caused by bacteria that can damage bone. Raman handheld spectroscopy has emerged as a promising diagnostic tool for detecting bone infection and can be used intraoperatively during surgical procedures. This study involved 120 bone samples from 40 patients, with 80 samples infected with either *Staphylococcus aureus* or *Staphylococcus epidermidis.* Raman handheld spectroscopy demonstrated successful differentiation between healthy and infected bone samples and between the two types of bacterial pathogens. Raman handheld spectroscopy appears to be a promising diagnostic tool in bone infection and holds the potential to overcome many of the shortcomings of traditional diagnostic procedures. Further research, however, is required to confirm its diagnostic capabilities and consider other factors, such as the limit of pathogen detection and optimal calibration standards.

## 1. Introduction

The technique of bone grafting is frequently utilized in orthopedic surgery to encourage spinal fusion and reconstruct bone defects resulting from a range of factors, including trauma, tumors, or revision arthroplasty [1,2,3,4,5,6,7,8]. Human bone allografts are frequently used to repair skeletal defects, as these induce bone remodeling and prevent early implant subsidence. In contrast to bone autografts, the supply of bone allografts is less limited, and donor site morbidity is not an issue with allografts. Particulate bone allografts are also increasingly used to address bone loss resulting from revision hip arthroplasty procedures. According to soil mechanics, well-graded, rigidly contained aggregates are recommended for firmly compacted bone grafts [6,9,10,11,12,13]. Bone allografts are harvested from the same species and preserved at −80 °C for up to five years [14]. Bone banks screen graft material rigorously for infectious pathogens, and bone transplants are treated with antibiotics, washed, and stored at −80 °C to ensure safety and effectiveness [6]. Bone grafts provide mechanical support and can also act as a local source of antibiotic treatment [5,15,16]. Detecting colonization bacteria and searching for concealed biofilms on allografts is crucial to prevent contamination and biofilm development during bone grafting. 

Bone infection, such as periprosthetic joint infection (PJI) and fracture-related infection (FRI), is another field where more rapid and accurate diagnostic tools for identifying causative pathogens are desired. PJI and FRI continue to be one of the most common and serious complications in musculoskeletal surgery and represent a major concern for patients, surgeons, and healthcare providers. These infections can result in high levels of disability and even death, while putting a significant financial burden on the healthcare system [17,18]. The orthopedic community is increasingly alarmed about managing implant-related infections due to the projected increase in patients experiencing this issue [19]. Treatment of these infections is challenging and often necessitates multiple surgical interventions [20,21]. PJI, for example, is a devastating condition, with fatality rates comparable to those of breast cancer and melanoma [8]. Coagulase-negative Staphylococci (CoNS), particularly *Staphylococcus epidermidis*, and then *Staphylococcus aureus* and mixed flora are the most common microbes in bone- and implant-related infections [14,22,23]. Unfortunately, these microbes have become increasingly resistant to commonly used antibiotics, such as penicillin, oxacillin, ciprofloxacin, clindamycin, erythromycin, and gentamicin, rendering these drugs significantly less effective [24]. Furthermore, the fact that these bacteria can form biofilms contributes to this phenomenon and negatively affects CoNS’s antimicrobial susceptibility [25]. Biofilm development also explains why some common flora species previously considered “harmless” can become infectious when they settle on the surface of foreign objects. 

Laboratory spectroscopy using mid-infrared (MIR) has proven to be an effective tool in diagnosing various bone diseases in humans [26,27,28,29,30,31,32,33,34] and determining bone quality [35]. MIR spectroscopy uses infrared radiation to interact with molecular vibrations within the 4000-to-400 cm^−1^ electromagnetic spectrum range. In this range, infrared radiation can stimulate molecular vibrations, like stretching, bending, and twisting modes, resulting in changes in the molecule’s dipole moment. These changes produce unique absorption bands that can be used to identify and quantify the sample’s molecular components [36,37].

Handheld spectroscopy has gained significant attention, alongside laboratory spectroscopy. The objective of this technique is to efficiently and precisely identify the properties of materials in a non-stationary manner. It encompasses a range of techniques that are highly effective in their applications. Handheld MIR spectrometers have several advantages over traditional benchtop spectrometers, such as being lightweight, compact, and battery powered. This enables an on-site analysis in various fields, requires minimal sample preparation, and provides rapid results [38].

Additionally, they are relatively affordable compared to benchtop spectrometers. However, they also have some limitations, such as lower spectral resolution, which limits their ability to distinguish between closely spaced absorption bands. Furthermore, they have a lower signal-to-noise ratio, leading to less accurate results. Additionally, handheld MIR spectrometers have a limited spectral range, hindering their ability to analyze complex samples [39].

A spectroscopic quality test must ensure the best bone graft samples for patients. This test can effectively determine the bone’s typical composition, including phosphate, carbonate bone mineral, collagen, and contaminants such as *Staphylococcus epidermidis* [34,40,41]. Experts must confidently identify and select only the highest-quality bone grafts through handheld MIR spectroscopy, which provides crucial information about mineralization processes and can efficiently detect contamination. Several studies in the last ten years have highlighted the usefulness of MIR spectroscopy for detecting different bone diseases in humans [26,27,28,29,30,31,32,33], including the evaluation of bone quality [35]. Bone is composed of various Raman bands, including phosphate (ν_3_PO_4_^3−^), carbonate (ν_1_CO_3_^2−^), collagen matrix, amide III, CH_2_ of protein, and amide I [33,42,43,44,45].

This study aimed to determine if Raman handheld spectroscopy can efficiently and reliably detect bacterial pathogens in human bone grafts, specifically *Staphylococcus aureus* and *Staphylococcus epidermidis*. We utilized freshly frozen bone samples obtained from hip replacement surgery patients and evaluated the suitability of a portable Raman spectrometer in tracking quality characteristics in bone samples. The study also explored the practicality of various methods, such as spectral evaluation, evaluation based on bone-specific parameters, or principal component analysis, for potential application during surgical procedures in future studies. The study highlights the ability of Raman handheld spectrometers to detect pathogens in human bone grafts and to aid in selecting and categorizing high-quality bone samples. 

## 2. Results

The focus of this experimental study was to distinguish infected human bone samples from healthy bone samples, using a handheld Raman device; and, subsequently, to differentiate between the two primary pathogens of osteomyelitis, *Staphylococcus aureus* and *Staphylococcus epidermidis*, in the infected bone samples. In total, 120 bone samples from 40 patients were examined.

### 2.1. Spectroscopy Data Evaluation

Through an analysis of the Raman spectra, we gained valuable insight into the chemical composition of the bone tissue under examination. In Figure 1, we observe the spectra from five spectra recorded from five different positions of a bone sample devoid of any infection, one contaminated with *Staphylococcus aureus*, and one infected with *Staphylococcus epidermidis*—the most prevalent pathogens in osteomyelitis cases.

The prominent bands of healthy human bone (see Figure 1) include the following bone-specific parameters: phosphate (PO_4_^3−^), carbonate (CO_3_^2−^), proline and hydroxyproline matrix of collagen, phenylalanine, amide III, CH_2_ deformation of the protein (wagging), amide I, stretching of the C-H groups, and triacylglycerols.

The phosphate group is associated with mineral components and can be differentiated into four vibrational modes, ν_1_–ν_4_. The vibrational mode ν_1_ is between 960 and 961 cm^−1^, ν_2_ is between 420 and 450 cm^−1^, ν_3_ is between the ranges of 1035–1048 cm^−1^ and 1070–1075 cm^−1^, and the vibrational mode ν_4_ can be read between 587 and 604 cm^−1^ [42,43,44,46,47,48,49,50,51,52]. The internal vibrational modes for the bone mineral carbonate (CO_3_^2−^) can be detected at a wavelength of 1070 cm^−1^ for the type B carbonate and 1103 cm^−1^ for the type A carbonate [49].

The Raman bands of the organic components are located at 851, 873, and 917 cm^−1^ for the proline and hydroxyproline matrix of collagen, while a maximum band at 1001 cm^−1^ is characteristic of the amino acid residue [43,44,46]. The peak for amide III is typically between 1200 and 1320 cm^−1^, and for amide I, the peak is in the wavelength range between 1595 and 1700 cm^−1^. A peak at 1450 cm^−1^ is indicative of CH_2_ deformation (wagging). A band between the ranges of 1400–1470 cm^−1^ and 2800–3000 cm^−1^ in the Raman spectrum can be attributed to the stretching of the C-H groups [45,49,51]. The spectral analysis also revealed a peak at 1748 cm^−1^, which corresponds to the C=O stretching vibrations of Triacylglycerols (TAGs) [53]. The presence of adipose tissue was detected, indicating that it was not fully removed during the cleaning process. Adipose tissue comprises lipids—mainly TAGs [54]—and can also be found in tendons [55,56,57]. This suggests that the extracted sample may have contained residual adipose tissue, which should be considered.

However, the Raman spectrum of the Mira handheld only extends to a wavelength of 2300 cm^−1^, which is why detecting a C-H stretching at 2800–3000 cm^−1^ is not feasible with this device. 

Specific changes can be observed when comparing the average spectrum of a bone sample without any inoculation to the Raman spectrum of a bone sample with *Staphylococcus aureus* inoculation. The ν_2_ phosphate band at 420–450 cm^−1^ and the ν_1_ vibration at 960 cm^−1^ decrease in intensity. However, the ν_3_ and ν_4_ phosphate bands and the bone mineral carbonate band only slightly decrease. The two-peaked band of amide III changes, with the second peak being higher than the first in the infected bone Raman spectrum. The C-H stretching at 1400 cm^−1^ increases in expression with *Staphylococcus aureus* infection. However, the CH_2_ deformation (wagging) at 1450 cm^−1^ and the amide I band at 1595–1700 cm^−1^ have reduced peak heights.

When comparing the Raman spectra of a bone chip without inoculation to one inoculated with *Staphylococcus epidermidis* (Figure 1), almost identical conditions in terms of the ν_3_ and ν_4_ oscillations of the phosphate can be observed, similar to the comparison made earlier with *Staphylococcus aureus*. However, the two peaks of amide III are reversed, with a lower absolute peak height than the sample with *Staphylococcus aureus*. The ν_2_ and ν_1_ oscillations of the phosphate and the carbonate band show a more prominent expression in the sample inoculated with *Staphylococcus epidermidis* than in the sample inoculated with *Staphylococcus aureus*. Notably, this comparison shows a significant difference in the reduction in the CH_2_ deformation at 1450 cm^−1^ and the amide I band at 1595–1700 cm^−1^.

The consistent location of infection changes in bone samples inoculated with staphylococci to simulate osteomyelitis is undeniable. However, relying solely on an optical analysis of Raman spectra may not be sufficient. Therefore, statistical methods are utilized to calculate band ratios and obtain additional information.

The data collected with the Mira Raman handheld device were analyzed using an Excel spreadsheet created for this study. The intensity and peak areas were determined from these data, and a one-factor ANOVA analysis was conducted, as shown in Figure 2. Table 1 summarizes the crucial bone-specific parameters examined in the present Raman spectra, which provide basic information about the quality and composition of a bone sample. The most influential bands’ intensities (I) resulted in the bands and band ratios listed, providing critical information for diagnosing a bone infection.

Significant differences exist between a healthy bone sample and one infected with *Staphylococcus aureus* in various parameters, including phosphate, amide I, mineral–matrix ratio (ratio of phosphate to amide I), and mineral carbonate content (MinCarb). In contrast. Comparing an uninoculated sample to one inoculated with *Staphylococcus epidermidis* reveals nearly indistinguishable results, except for a slight variation in the calculation of mineral carbonate content. However, comparing bone samples infected with *Staphylococcus aureus* and *Staphylococcus epidermidis* highlights a considerable difference in mineral carbonate content.

The amide I band is an indicator of protein crosslinking and protein conformation. Changes in the collagen network during an infection lead to a reduced structural organization and, thus, weaker bone strength [58]. The ratio of phosphate to amide I, known as the mineral–matrix ratio (MMR), measures the degree of mineralization in bone tissue and is thus used to assess bone quality and strength [34,46,59,60]. The results of this experimental study show an increased loss of relative mineral content in bone in the context of bacterial infection, affecting the bone tissue’s strength, making it more susceptible to fracture in contrast to healthy bone tissue [61,62].

The Raman spectra parameters analysis by one-factorial ANOVA shows that differentiation between healthy human bone tissue and infected bone is possible. Differentiation between the two primary pathogens of osteomyelitis, *Staphylococcus aureus* and *Staphylococcus epidermidis*, is only possible regarding mineral carbonate content (MinCarb). The co-cultivation of *Staphylococcus aureus* and *Staphylococcus epidermidis* caused a significant deterioration of bone quality and protein conformation. No detection of specific correlations is possible with this processing method, so a principal component analysis (PCA) was also performed to characterize the range of spectral variations.

### 2.2. Diagnostic Performance PCA

PCA is an essential diagnostic tool for making quick and accurate treatment decisions [63]. In this experimental study, PCA was utilized to visualize changes in bone tissue by analyzing the mean spectra of 120 bone samples. The first principal component is based on the Raman spectra of 40 bone chips that were not inoculated. The second principal component consists of 40 bone chips previously inoculated with *Staphylococcus aureus*, and the third principal component comprises 40 samples with the bacterium *Staphylococcus epidermidis*. It is imperative to utilize PCA in diagnostic procedures to ensure accurate and efficient results in bone tissue analysis. During the first stage, bone samples were compared using Raman spectroscopy to determine if it was possible to differentiate between healthy and infected bone tissue. A “loadings plot” should be employed to identify the principal components and highlight the main variance in the dataset, which calculates and displays the correlation between individual variables. This plot consists of correlation coefficients, with a maximum value of 1 and a minimum value of −1, and is depicted graphically (refer to Figure 3). Values that are identical when comparing the three types of bone samples are represented by the zero line. Deviations in the positive or negative range allow for differentiation between the three principal components. It has been shown through this experimental study that such differentiation is possible.

The detailed results of the PCA spectral analysis are shown in Figure 4 and Table 2. Table 2 shows that comparing uninoculated and infected bone samples using a principal component analysis yields results that are almost identical to the statistical results. In the context of a PCA, only the first two principal components, PC-1 and PC-2, are usually considered since these already contain 95% of the investigated information. The percentage distribution of the principal components for the respective 3D score plots can be read from column 2 of Table 2. As already described, the parameters phosphate, the ratio of phosphate to amide I (mineral–matrix ratio), the mineral carbonate content (MinCarb), and amide I are significantly different when comparing a healthy with an infected human bone sample. From Figure 4, as well as Table 2, it can be seen that the 3D score plots numbered III, VI, and VII show the most significant results. Here, 3D score plot III is also due to the bone parameter phosphate at 958 cm^−1^; 3D score plot VI describes the CH_2_ stretching (wagging) at 1450 cm^−1^, which is used to calculate the mineral carbonate content (MinCarb); and 3D score plot VII represents the amide I band at 1656 cm^−1^, from which the mineral–matrix ratio (MMR) can subsequently be calculated. Thus, principal component analysis (PCA), as the most accurate method to represent the correlation of bone-specific parameters, underscores the results obtained in this experimental study.

In the second step, finding whether a pathogen-specific differentiation of the inoculated bone samples was also possible was necessary. For this purpose, the 80 bone samples previously inoculated with either *Staphylococcus aureus* or *Staphylococcus epidermidis* were compared using a second PCA. The results of this analysis are shown in Figure 5 and Table 3.

Section 2.1 demonstrates that mineral carbonate content (MinCarb) can effectively distinguish between bone samples inoculated with *Staphylococcus aureus* and *Staphylococcus epidermidis*, using one-factorial ANOVA. However, the principal component analysis (PCA) results show even more meaningful results, as seen in Figure 5 and Table 3. The 3D score plots III, VI, and VII are particularly useful. The bone parameters phosphate (III), the CH_2_ stretch (VI), and the amide I band (VII) provide 93% of the information from the first main component PC-1, with 4% to 6% falling back to the second principal component PC-2. Therefore, PCA can detect changes in bone tissue in bacterial osteomyelitis, and Raman handheld spectroscopy can enable immediate differentiation between *Staphylococcus aureus* and *Staphylococcus epidermidis* even during surgery.

## 3. Discussion

To distinguish between uninfected and infected bone samples inoculated with the two most common pathogens of osteomyelitis, *Staphylococcus aureus* and *Staphylococcus epidermidis*, a handheld Raman instrument and spectral analysis and principal component analysis (PCA) were used in this experimental study. The present results demonstrate that Raman spectroscopy can provide meaningful results using recommended sample preparation, measurement settings, and data analysis strategies. Human bone samples co-cultured with *Staphylococcus aureus* and *Staphylococcus epidermidis* exhibited significant loss of bone quality and protein conformation. Using the carbonate/phosphate ratio, the mineral–matrix ratio determined that the relative mineral content decreased more in infected human bone than in uninfected bone. In addition, the amide I band was used to detect infection-related changes in the collagen network. The principal component analysis was used to identify *Staphylococcus aureus* and *Staphylococcus epidermidis* in different spectral regions, primarily in the bending and stretching modes of phosphate, amide I, and C-H groups. Therefore, our results indicate that handheld Raman spectroscopy can distinguish between infected and non-infected bone samples and differentiate between the two primary pathogens of osteomyelitis, *Staphylococcus aureus* and *Staphylococcus epidermidis*.

The current gold standard for diagnosing periprosthetic joint infections is the examination of cultures of tissue samples obtained during the surgical procedure. This approach is resource-intensive and time-consuming, with results available no earlier than five-to-eleven days after specimen collection [64]. Another problem is posed by false-positive analytical results due to contamination with other pathogens resulting from inadequate processing and poor transport conditions to the laboratory [65]. Additional histopathological examination requires large amounts of sample material, and, at the same time, the sensitivity and specificity of this diagnostic method are lower [66,67]. In contrast, Raman spectroscopy requires only a tiny bone sample that can be examined in situ during a surgical procedure. The results are available directly in the operating room, without delay, and are highly accurate. Raman spectroscopy is a practical and convenient approach to tracking multiple independent chemical and biochemical changes in biological samples. 

Direct detection of bone infection using Raman spectroscopy is challenging due to the complex structure of bone tissue, low pathogen concentrations, and spectral overlap. However, Raman spectroscopy is a fast, automatable, and robust technique with numerous advantages. It requires only a tiny biological sample, making this technique ideal for situations with only a limited amount of tissue. Thus, this technique is particularly advantageous when treating patients with osteomyelitis is urgent because the analytical results are available without significant time delay. The current gold standard for diagnosing periprosthetic joint infections is the culture of intraoperatively harvested tissue samples, which is a very resource- and time-intensive procedure. In addition, results are available, at the earliest, five-to-eleven days after intraoperative collection [64], and false-positive analysis results may be a problem due to contamination with other pathogens resulting from inadequate processing and poor transport conditions on the way to the laboratory [65]. An additional histopathological examination requires relatively large amounts of tissue, and the sensitivity and specificity of this diagnostic method are even lower [66,67]. 

However, our research has certain limitations, including the limited availability of bone samples from our local bone bank and the short incubation time of bone fragments to promote bacterial growth. In addition, the study was unblind, and it was impossible to determine the limit of detection (LOD) of the pathogen. This warrants further study to help establish calibration standards with known pathogen concentrations. The LOD describes the minor possible pathogen concentration that can be distinguished from the signal-to-noise ratio (SNR) with a certain probability in Raman spectroscopy. It depends on various factors, such as the instrument configuration, the sample properties, the analytical method chosen, and the properties of the exciter. However, direct pathogen detection is not always possible, despite specific spectral changes. Therefore, optimal sample preparation and the optimization of instrument parameters and data analysis are essential to define the LOD more accurately in the future. In addition, it is advisable to obtain detailed information on the instrument used from the manufacturer or an expert.

The results of our experimental study highlight the potential of spectroscopic analyses to link molecular changes to pathological changes. We found that handheld Raman spectroscopy can detect bacterial infections in human bone samples and may hold the potential to overcome the essential shortcomings of currently used microbiological and histopathological diagnostic procedures. However, further studies with larger sample sizes and a greater variety of bacterial pathogens are needed to uncover potential confounding variables and validate this technique as a new diagnostic tool for inflammatory bone diseases. 

## 4. Materials and Methods

### 4.1. Sample Collection

The femoral heads stored at our bone bank come from individuals who underwent hip replacement surgery due to advanced hip osteoarthritis or femoral neck fracture. Before donating their bones, patients give their written consent, indicating their willingness to contribute and their complete understanding of the process. We keep any donated bone that does not meet therapeutic criteria, such as incomplete screening and documentation, and use it for scientific research. We do not collect bone from individuals with severe osteoporosis or contaminated samples from various pathogens, regardless of research needs. Age and gender are not criteria for selecting donors. We cool and drain the bones with 0.9% saline during osteotomy to prevent heat damage. We remove cartilage and cortical tissues from the femoral heads with a bone saw. We then extract bone chips that are 3–5 mm in diameter from the residual spongious tissue, using a bone mill (Noviumagus Bone Mill; Spierings Meische Techniek BV, Nijmegen, The Netherlands). We examined 40 human bone samples, and the local ethics council approved our retrospective study (EK 1291/2021) based on the guiding principles stated in the Declaration of Helsinki.

### 4.2. Development of Biofilm on Bone Allografts

A total of 120 bone chips from 40 patients were used to perform this experimental study (see Table 4). Of these, 40 bone chips were inoculated with *Staphylococcus aureus* (type ATCC 29213) and *Staphylococcus epidermidis* (type ATCC 12228), while the remaining 40 bone samples remained uninoculated. Bacterial culture incubation was performed in Mueller–Hinton medium at 37 °C for 24 h. Subsequently, each inoculum was diluted to 106 CFU/mL, and 200 µL of this diluted suspension was transferred into each multi-well plate. Bone samples were added individually with a substrate to form a biofilm and incubated in an orbital shaker (Edmund Bühler GmbH, Bodelshausen, Baden-Württemberg, Germany) for 48 h at 37 °C. To allow for successful biofilm formation, incubation was performed in a humid chamber designed to illustrate the contamination of bone tissue. After the incubation, the remaining suspension was removed, and the bone chips were washed with PBS (phosphate-buffered saline). The bone samples were then dried in an aspirator (3.2 kPa) for 10 min at room temperature and measured. It is worth noting that prolonging the drying time to 24 h did not cause differences in spectra quality. Hence, the 10-min drying period was sufficient.

### 4.3. MIRA Raman Handheld

Raman spectra were obtained using the Mira Raman handheld (Metrohm Inula GmbH, Wien, Österreich). The measurements were conducted using orbital raster scanning in a circular mode, with a spot size of 42 µm, and the measured area was 0.332 mm^2^. The polarization degree was 1000:1, and the device operated at a wavelength of 785 nm. The spectral range covered was from 2300 to 400 cm^−1^, with a spectral resolution of 6 cm^−1^. Each sample was subjected to five spectra recorded from five different positions. The measurement was conducted at 22 °C, under controlled humidity levels.

### 4.4. Data Processing

The Unscrambler X 10.5 (AspenTech, Bedford, MA, USA) was used for data processing, including a reduction factor of 36, 15-point Savitzky–Golay smoothing, and area normalization. Comparing peak areas of different samples, such as I958 or amide-I in Figure 3, required area normalization. This was especially important for porous samples like bones, where the excitation volume is not uniform, and the beam is not always well focused [68]. Frequency shifts and background subtraction were not performed and were necessary. To study diagnostic parameters, peak intensity (I) [69,70,71,72,73,74] was analyzed with the help of an Excel spreadsheet that manages both spectroscopic and chromatographic data [75]. Subsequent to selecting the desired peak from the “Output” section, the baseline was adjusted by subtracting a linear equation derived from the x1:y1 and xn:yn values. The maximum height (H) and peak area from there can be determined. Two area values are available: one is calculated through Equation (1) by taking the partial sum of the peak areas (A), while the other is calculated by summing all intensities (Asi) via Equation (2) [75].
(1)$$A=ai+aj+\ldots +an,\;being\;an=\frac{\left(\mathrm{yn}-1+\mathrm{yn}\right)\times \left(xn-1-xn\right)}{2}$$
(2)A=y1+y2+…+yn

A statistical analysis of spectral parameters was conducted using GraphPad Prism software (version 9, San Diego, CA, USA) and compared through a two-sample *t*-test. A significant result is only considered if the *p*-value is less than 0.05.

### 4.5. Principal Component Analyses (PCA)

Unscrambler X 10.5 was used to create PCA models by importing spectra and applying data pretreatments, including a reduction factor of 36, 15-point Savitzky–Golay smoothing, and area normalization.

## 5. Conclusions

Raman spectroscopy helps track chemical and biochemical changes in biological samples. However, detecting bone infections is challenging due to the complexity of bone tissue and low pathogen concentrations. To overcome this, we used a handheld Raman device with spectral and principal component analyses to differentiate between infected and uninfected bone samples and distinguish between the two most common bacterial pathogens of bone infection. Our study found that infected bones exhibited significant deterioration and protein deformation, resulting in a decline in relative mineral content and changes in the collagen network. Raman spectroscopy has excellent potential as a novel diagnostic tool for bone infection. In contrast to traditional diagnostic procedures, it yields results without delay, can be applied even in the operating theater, and requires only a minimal amount of bone sample. Nevertheless, further investigations are necessary to elucidate its potential as a diagnostic tool for inflammatory bone conditions.

## Figures and Tables

**Figure 1 ijms-25-00541-f001:**
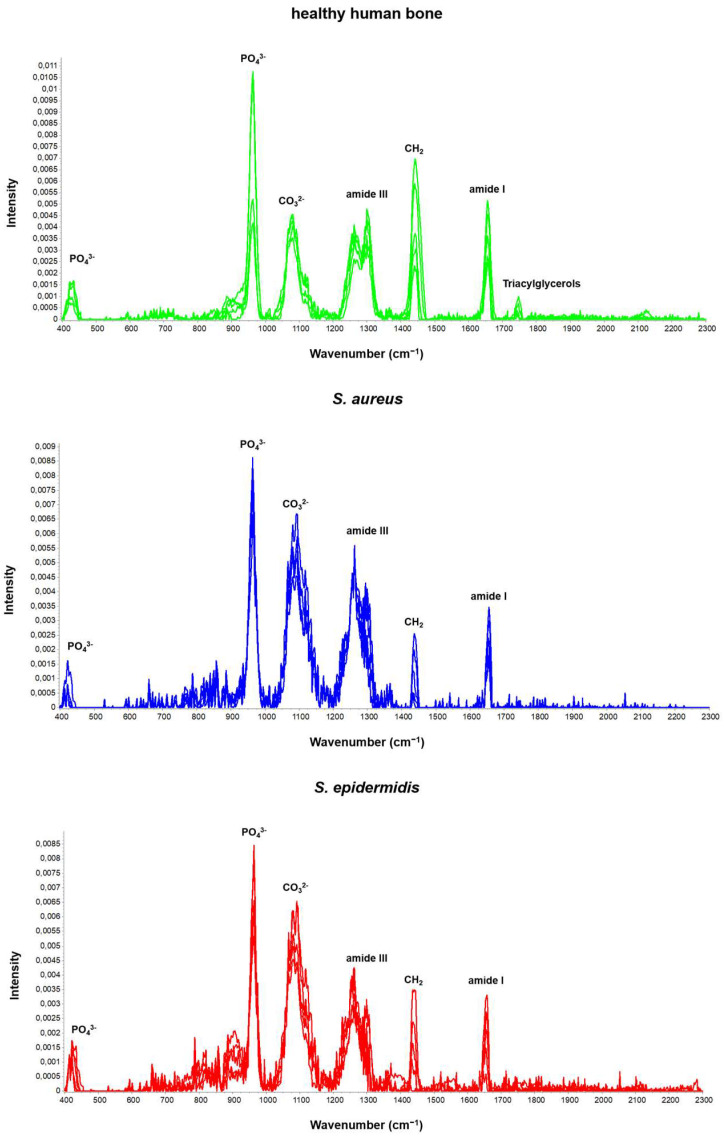
Representative Raman spectrum of an uninoculated bone sample and two inoculated bone samples (*Staphylococcus aureus* and *Staphylococcus epidermidis*).

**Figure 2 ijms-25-00541-f002:**
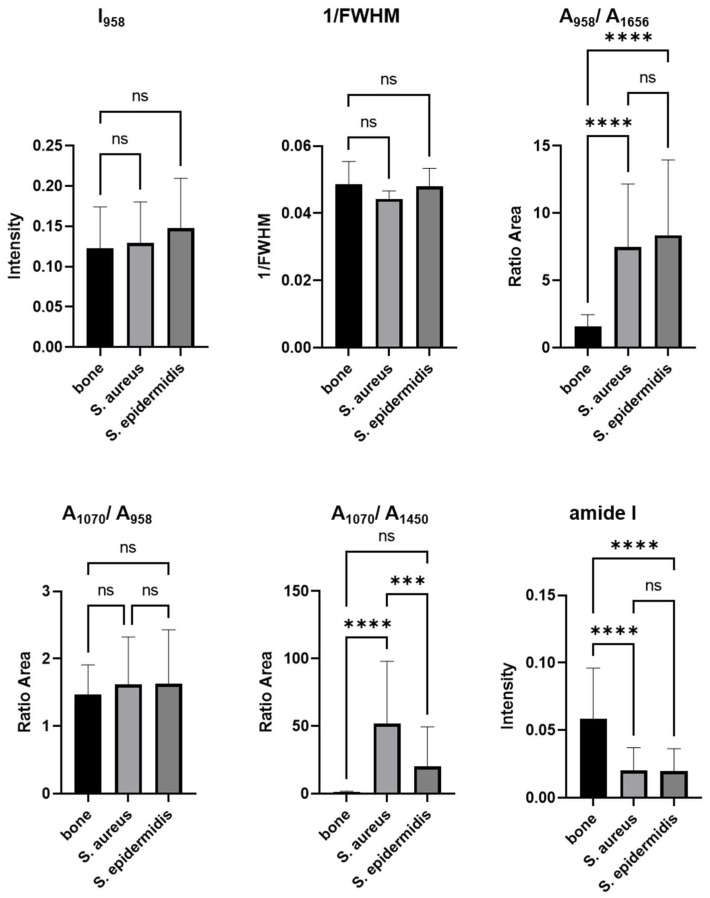
Illustration of Mira Raman handheld-derived spectral markers for human bone characterization, using a bar graph. *** Significant; **** highly significant; ns, not significant.

**Figure 3 ijms-25-00541-f003:**
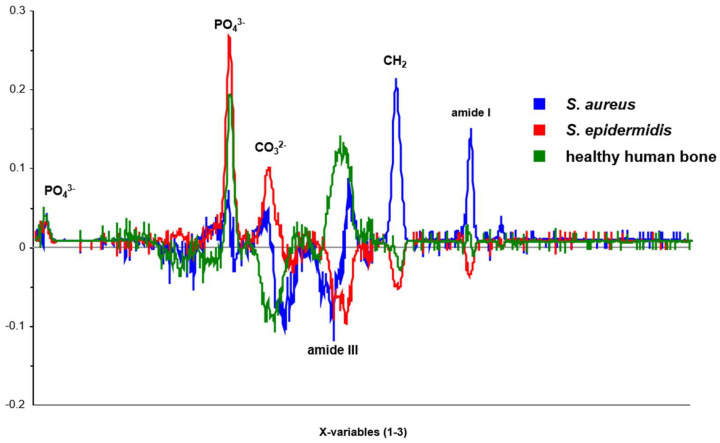
Loadings plot for the main components: PC-1, PC-2, and PC-3.

**Figure 4 ijms-25-00541-f004:**
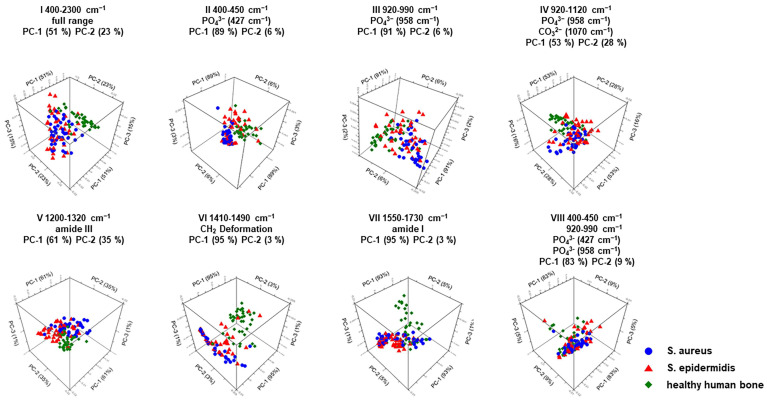
PCA for 3D score plots of Mira’s handheld Raman instrument. I to VIII are the score plots between PC-1 and PC-2 for uninfected and infected human bone.

**Figure 5 ijms-25-00541-f005:**
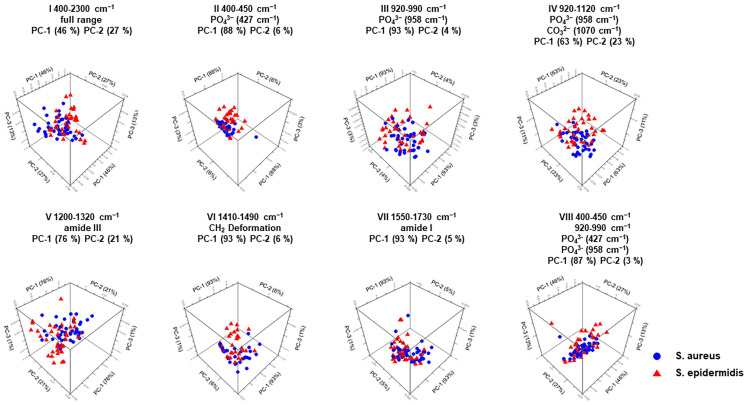
PCA for 3D score plots of the Mira handheld Raman instrument. I to VIII are the score plots between PC-1 and PC-2 for human bone infected with *Staphylococcus aureus* and *Staphylococcus epidermidis*.

**Table 1 ijms-25-00541-t001:** Mira Raman handheld-derived spectral markers for human bone characterization (I, band intensity; A, band area; and FWHM, full width at half maximum) and diagnostic significance (*p* value) based on ordinary one-factorial ANOVA. The *p*-values < 0.05 are considered significant; *** significant; **** highly significant; ns, not significant.

Name	Description	Determination	*p*-Values Based on a One-Factor ANOVA		
			Healthy Human Bone	*Staphylococcus epidermidis*	*Staphylococcus aureus*
Phosphate	ν_3_PO_4_^3−^ Amount of phosphate	(I_958_)	<0.0001 ****	<0.0001 ****	0.9946 ^ns^
Crystallinity		1/FWHM_958_	0.1572 ^ns^	0.9431 ^ns^	0.2728 ^ns^
Mineral/matrix (MMR) phosphate/amide I	ν_3_PO_4_^3−^/amide I Mineral component amount to the organic one	(A_958_/A_1656_)	<0.0001 ****	<0.0001 ****	0.7055 ^ns^
Mineral quality and crystallinity carbonate/phosphate	ν_1_CO_3_^2−^/ν_1_PO_4_^3−^ Carbonate incorporation extent in the hydroxyapatite lattice	(A_1070_/A_958_)	0.6208 ^ns^	0.5697 ^ns^	0.9968 ^ns^
Mineral carbonate content (MinCarb)	ν_1_CO_3_^2−^/(C-H) bend; CH_2_ wag	(A_1070_/A_1450_)	<0.0001 ****	0.0943 ^ns^	0.0007 ***
Amide I	Amide I of α-helical structures Arrangement and quantity of collagen	(I_1656_)	<0.0001 ****	<0.0001 ****	0.9946 ^ns^

**Table 2 ijms-25-00541-t002:** PCA of uninfected human bone and bone infected with *Staphylococcus aureus* and *Staphylococcus epidermidis* measured with Mira Raman handheld.

Wavenumber Range Number	PCA	Assignment	Spectral Region
I	PC-1 (51%) PC-2 (23%)	Full wavenumber range	400 cm^−1^ to 2300 cm^−1^
II	PC-1 (89%) PC-2 (6%)	PO_4_^3−^ (427 cm^−1^)	400 cm^−1^ to 450 cm^−1^
III	PC-1 (91%) PC-2 (6%)	PO_4_^3−^ (958 cm^−1^)	920 cm^−1^ to 990 cm^−1^
IV	PC-1 (53%) PC-2 (28%)	PO_4_^3−^ (958 cm^−1^) CO_3_^2−^ (1070 cm^−1^)	920 cm^−1^ to 1120 cm^−1^
V	PC-1 (61%) PC-2 (35%)	Amide III (1246 cm^−1^)	1200 cm^−1^ to 1320 cm^−1^
VI	PC-1 (95%) PC-2 (3%)	CH_2_ deformation (1450 cm^−1^)	1410 cm^−1^ to 1490 cm^−1^
VII	PC-1 (93%) PC-2 (5%)	Amide I (1656 cm^−1^)	1550 cm^−1^ to 1730 cm^−1^
VIII	PC-1 (83%) PC-2 (9%)	PO_4_^3−^ (427 cm^−1^) PO_4_^3−^ (958 cm^−1^)	400 cm^−1^ to 450 cm^−1^ 920 cm^−1^ to 990 cm^−1^

**Table 3 ijms-25-00541-t003:** PCA of human bone infected with *Staphylococcus aureus* and *Staphylococcus epidermidis* measured with Mira Raman Handheld.

Wavenumber Range Number	PCA	Assignment	Spectral Region
I	PC-1 (46%) PC-2 (27%)	Full wavenumber range	400 cm^−1^ to 2300 cm^−1^
II	PC-1 (88%) PC-2 (6%)	PO_4_^3−^ (427 cm^−1^)	400 cm^−1^ to 450 cm^−1^
III	PC-1 (93%) PC-2 (4%)	PO_4_^3−^ (958 cm^−1^)	920 cm^−1^ to 990 cm^−1^
IV	PC-1 (63%) PC-2 (23%)	PO_4_^3−^ (958 cm^−1^) CO_3_^2−^ (1070 cm^−1^)	920 cm^−1^ to 1120 cm^−1^
V	PC-1 (76%) PC-2 (21%)	Amide III (1246 cm^−1^)	1200 cm^−1^ to 1320 cm^−1^
VI	PC-1 (93%) PC-2 (6%)	CH_2_ deformation (1450 cm^−1^)	1410 cm^−1^ to 1490 cm^−1^
VII	PC-1 (93%) PC-2 (5%)	Amide I (1656 cm^−1^)	1550 cm^−1^ to 1730 cm^−1^
VIII	PC-1 (87%) PC-2 (3%)	PO_4_^3−^ (427 cm^−1^) PO_4_^3−^ (958 cm^−1^)	400 cm^−1^ to 450 cm^−1^ 920 cm^−1^ to 990 cm^−1^

**Table 4 ijms-25-00541-t004:** Sample properties.

Age	Female	Male
<50	2	3
50–60	6	2
60–70	4	5
70–80	9	7
>80	1	1

## Data Availability

The data presented in this study are available upon request from the corresponding author.

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
