# Peer review of "Enhancing Bone Infection Diagnosis with Raman Handheld Spectroscopy: Pathogen Discrimination and Diagnostic Potential"

_ijms, 2023, doi:10.3390/ijms25010541_

Round 1
Reviewer 1 Report
Comments and Suggestions for Authors
The paper deals with the application of Raman spectroscopy for the evaluation of bone quality and the determination of bone alterations upon inoculation with Staphylococci. The application of spectroscopic techniques during surgical operations is very promising as it provides useful indicators for alterations occurring even at the molecular level that cannot be visibly observed. Regarding the originality, a similar study with similar results has been published in 2018 (doi: 10.1038/s41598-018-27752-z) which is cited as Ref. 33.
The paper can be published with minor revision.
Comments
1. L61 “Staphylococcus aureus”: convert to italics
2. L99 “MIR bands”: Replace with “Raman bands” as these features are also detectable with visible light lasers.
3. L129, Figure 1 & L138, Figure 2: To my opinion Figure 1 is not necessary as the related information is also provided in the text. This figure could serve as a graphical abstract if it is requested by the journal. On the other hand, Figure 2 contains useful information and should be expanded (the band labels can be barely seen in the printed manuscript). Please increase the number of “ticks” in the horizontal scale (e.g. every 20 or 50 cm-1). The inclusion of a graph that will show the 5 different spectra recorded from different spots on the same sample would be a useful addition.
4. L133 & L138, Figure 2: In the text it is stated that the spectra shown in Fig. 2 are “average spectra”. What does average mean? Does it correspond to the average of more than one acquisition from the same sample or it is the average of the spectra of all the healthy samples? Please clarify.
5. L133, Figure 2: Please comment on the peak that appears after the amide I peak in the spectrum of the healthy bone. Does it correspond to lipids (doi: 10.1002/jrs.4607, 10.1016/j.chroma.2017.08.004, 10.1111/febs.14506)? If yes, is there a possibility that other peaks from the lipid spectrum interfere with other bone components?
6. L153 “phenylalanine parameter”: Replace parameter with amino acid residue.
7. L171: covert 3 and 4 in v3 and v4 to subscripts
8. L189, Figure 3: Please mention the way you determined the area under the peaks/bands. Did you fit the spectra? If yes, which kind of functions did you use? Did you determine the area with integration?
9. Figures 5 & 6: Increase the font size in 3D representations.
10. Table 2 & 3: replace “bis” with “to”.
12. L372 “to 0 cm-1”: 0 cm-1 is usually not detectable as Rayleigh elastic scattering is very intense and could destroy the detector.
13. L371: Please provide the spot size of the laser beam and its polarization state (polarization degree). Did you check for polarization/angular dependence of the spectra (acquisition of spectra at different angles of incidence of the beam to the sample surface and after rotating the sample around an axis). Please comment on that (see for example doi: 10.1117/1.3426310, 10.1016/j.actbio.2020.10.034, 10.1039/D1AN01560E).
14. L377: Please mention if and how the spectra were normalized, whether they are corrected for frequency shifts and if the spectra were subjected to background subtraction. A normalization procedure is necessary when comparing peak areas of different samples (e.g. I958 or amide-I in Figure 3), especially in the case of porous samples like bones where is excitation volume is not always the same and when the beam is not well focused (doi: 10.1039/D1AN01560E)
Author Response
Dear Editor, dear Reviewers,
Thank you for the constructive comments. We adapted the paper according to the comments made by the reviewers. In this response letter, we will document all our answers to the reviewers and state where the changes were applied in the final manuscript:
Rev 1
The paper deals with the application of Raman spectroscopy for the evaluation of bone quality and the determination of bone alterations upon inoculation with Staphylococci. The application of spectroscopic techniques during surgical operations is very promising as it provides useful indicators for alterations occurring even at the molecular level that cannot be visibly observed. Regarding the originality, a similar study with similar results has been published in 2018 (doi: 10.1038/s41598-018-27752-z) which is cited as Ref. 33.
The paper can be published with minor revision.
Comments
- L61 “Staphylococcus aureus”: convert to italics
AW: We converted “Staphylococcus aureus” to italics according to the reviewer’s comment.
- L99 “MIR bands”: Replace with “Raman bands” as these features are also detectable with visible light lasers.
AW: We replaced “MIR bands”: Replace with “Raman bands” according to the reviewer’s comment.
- L129, Figure 1 & L138, Figure 2: To my opinion Figure 1 is not necessary as the related information is also provided in the text. This figure could serve as a graphical abstract if it is requested by the journal. On the other hand, Figure 2 contains useful information and should be expanded (the band labels can be barely seen in the printed manuscript). Please increase the number of “ticks” in the horizontal scale (e.g. every 20 or 50 cm-1). The inclusion of a graph that will show the 5 different spectra recorded from different spots on the same sample would be a useful addition.
AW: We deleted Figure 1 according to the reviewer’s comment and will provide this image as a graphical abstract if requested by the journal. Additionally, as suggested, we increased the number of "ticks" in the horizontal scale to provide a more straightforward presentation.
Moreover, we agree that a graph showcasing the five different spectra recorded from various spots on the same sample is useful and incorporated this as a new figure in the revised version of the manuscript as proposed.
- L133 & L138, Figure 2: In the text it is stated that the spectra shown in Fig. 2 are “average spectra”. What does average mean? Does it correspond to the average of more than one acquisition from the same sample or it is the average of the spectra of all the healthy samples? Please clarify.
AW: We added the missing information into the text according to the reviewer’s comment.
- L133, Figure 2: Please comment on the peak that appears after the amide I peak in the spectrum of the healthy bone. Does it correspond to lipids (doi: 10.1002/jrs.4607, 10.1016/j.chroma.2017.08.004, 10.1111/febs.14506)? If yes, is there a possibility that other peaks from the lipid spectrum interfere with other bone components?
AW: Indeed, this peak is not fully explainable. We added the missing information based on the literature provided by the reviewer (doi: 10.1002/jrs.4607, 10.1016/j.chroma.2017.08.004, 10.1111/febs.14506): The spectral analysis also revealed a peak at 1748 cm-1, which corresponds to the C=O stretching vibrations of Triacylglycerols (TAGs) (54). The presence of adipose tissue was detected, indicating that it was not fully removed during the cleaning process. Adipose tissue comprises lipids, mainly TAGs(55) and can also be found in tendons (56-58). This suggests that the extracted sample may have contained residual adipose tissue, which should be considered.
- L153 “phenylalanine parameter”: Replace parameter with amino acid residue.
AW: According to the reviewer's comment, we replaced “phenylalanine parameter” by “amino acid residue”.
- L171: covert 3 and 4 in v3 and v4 to subscripts
AW: According to the reviewer's comment, we converted 3 and 4 in v3 and v4 to subscripts.
- L189, Figure 3: Please mention the way you determined the area under the peaks/bands. Did you fit the spectra? If yes, which kind of functions did you use? Did you determine the area with integration?
AW: We added the missing information according to the reviewer’s comment based on the publication (https://doi.org/10.1016/j.chemolab.2019.103816):
Subsequent to selecting the desired peak from the "Output" section, the baseline was adjusted by subtracting a linear equation derived from the x1:y1 and xn:yn values. The maximum height (H) and peak area from there can be determined. Two area values are available: one is calculated usiing Equation (A) by taking the partial sum of the peak areas (A), while the other is calculated by summing all intensities (Asi) via the following Equation (B) (70):
Equation A: A=ai+aj+⋯+an,being an=((yn-1+yn)*(xn-1-xn))/2
Equation B: A=y1+y2+⋯+yn
- Figures 5 & 6: Increase the font size in 3D representations.
AW: The font sizes in 3D representations were increased accordingly.
- Table 2 & 3: replace “bis” with “to”.
AW: According to the reviewer's comment, we replaced “bis” by “to” in Tables 2 and 3.
- L372 “to 0 cm-1”: 0 cm-1 is usually not detectable as Rayleigh elastic scattering is very intense and could destroy the detector.
AW: This was our mistake. We added the missing information according to the reviewer’s comment. The measurements were performed from 2300 to 400 cm-1.
- L371: Please provide the spot size of the laser beam and its polarization state (polarization degree). Did you check for polarization/angular dependence of the spectra (acquisition of spectra at different angles of incidence of the beam to the sample surface and after rotating the sample around an axis). Please comment on that (see for example doi: 10.1117/1.3426310, 10.1016/j.actbio.2020.10.034, 10.1039/D1AN01560E).
AW: We added the missing information according to the reviewer’s comment. Raman spectra were obtained using the Mira Raman handheld (Metrohm Inula GmbH, Wien, Österreich). The measurements were conducted using orbital raster scanning in a circular mode with a spot size of 42 µm and the measured area was 0.332 mm2. The polarization degree was 1000:1 and the device operated at a wavelength of 785 nm.
- L377: Please mention if and how the spectra were normalized, whether they are corrected for frequency shifts and if the spectra were subjected to background subtraction. A normalization procedure is necessary when comparing peak areas of different samples (e.g. I958 or amide-I in Figure 3), especially in the case of porous samples like bones where is excitation volume is not always the same and when the beam is not well focused (doi: 10.1039/D1AN01560E)
AW: We added the missing information according to the reviewer’s comment. The Unscrambler X 10.5 (AspenTech, Bedford, MA, USA) was used for data processing, including a reduction factor of 36, 15-point Savitzky-Golay smoothing, and area normalisation. Comparing peak areas of different samples, such as I958 or amide-I in Fig-ure 3, required area normalisation. This was especially important for porous samples like bones, where the excitation volume is not uniform, and the beam is not always well-focused (69). Frequency shifts and background subtraction were not performed and were not considered as necessary. This was added to the methods section.
Thanks again for the valuable comments! Kind regards,
Johannes Pallua, corresponding author
Priv.-Doz. MMag.Dr.rer.nat. Johannes Pallua MSc PhD
Univ.-Klinik für Orthopädie und Traumatologie
Anichstraße 35
A-6020 Innsbruck
Tel.: +43 50 504 80242
Mail: johannes.pallua@tirol-kliniken.at
Reviewer 2 Report
Comments and Suggestions for Authors
An interesting study and exploration of the possibilities for Raman. Some comments:
General Comments
- I think the authors need to be very careful about the interpretation of their results as the study feels a bit misleading. There was no information provided about the bacterial load on the evaluated bone samples and how this compares to what is clinically observed. This is a major limitation of the study, since it's difficult to know whether RAMAN would work at a lesser bacterial load. It should be emphasized (ideally in the title) that this is an ex vivo/experimental study.
- There was no blinding in this study which is also a major limitation and introduces bias into the interpretation of the results.
Introduction
- Sufficient as written.
Results
- Line 117-128 seem more fit for the Discussion (or Introduction) than the Results, as it is more focused on the motivation for why the study is important.
Discussion
- How do the authors think Raman would help to scan for polymicrobial infections?
Author Response
Dear Editor, dear Reviewers,
Thank you for the constructive comments. We adapted the paper according to the comments made by the reviewers. In this response letter, we will document all our answers to the reviewers and state where the changes were applied in the final manuscript:
Rev 2
An interesting study and exploration of the possibilities for Raman. Some comments:
General Comments
- I think the authors need to be very careful about the interpretation of their results as the study feels a bit misleading. There was no information provided about the bacterial load on the evaluated bone samples and how this compares to what is clinically observed. This is a major limitation of the study, since it's difficult to know whether RAMAN would work at a lesser bacterial load. It should be emphasized (ideally in the title) that this is an ex vivo/experimental study.
AW: This study aimed to investigate Raman as a technology for diagnosing PJI and identifying the pathogens causing the infection. Macroscopically, the difference coulod not be objectively assessed. Therefore this study showed the feasibility of an objective technology in detecting the presence of microorganisms in the bone samples. We agree that the quantification of contamination by calculating the load of microorganisms has to be considered in future studies, but was not the aim at this phase of the study. We agree with the idea of displaying “ex-vivo” or even “proof-of-concept” on the title and are considering a revision.
- There was no blinding in this study, which is also a major limitation and introduces bias into the interpretation of the results.
AW: We agree, but for this validation study, we did not use blinding of the samples. However, we agree that this is important for any further studies after this proof-of-concept.
Introduction
- Sufficient as written.
Results
- Line 117-128 seem more fit for the Discussion (or Introduction) than the Results, as it is more focused on the motivation for why the study is important.
AW: We transferred the text to the discussion section according to the reviewer's comment.
Discussion
- How do the authors think Raman would help to scan for polymicrobial infections?
AW: We added the missing information according to the reviewer’s comment: The unique spectral fingerprint of various bacterial and fungal species could be utilized to differentiate them from each other and monitor the changes in the microbial community over time. This approach could be beneficial in clinical settings for diagnosing infections and selecting appropriate treatment strategies.
Thanks again for the valuable comments! Kind regards,
Johannes Pallua, corresponding author
Priv.-Doz. MMag.Dr.rer.nat. Johannes Pallua MSc PhD
Univ.-Klinik für Orthopädie und Traumatologie
Anichstraße 35
A-6020 Innsbruck
Tel.: +43 50 504 80242
Mail: johannes.pallua@tirol-kliniken.at
Round 2
Reviewer 2 Report
Comments and Suggestions for Authors
Thank you for your additions to this manuscript.
Comments on the Quality of English LanguageLine 319: "inblinded" should be "unblinded" or similar.
Author Response
Dear Editor, dear Reviewer 2,
Thank you for the constructive comment. We adapted the paper according to the comment made by the Reviewer 2. In this response letter, we will document all our answers to the reviewers and state where the changes were applied in the final manuscript:
Reviewer 2
Line 319: "inblinded" should be "unblinded" or similar.
AW: We replaced "inblinded" with unblinded" according to the reviewer’s comment.
Thanks again for the valuable comment! Kind regards,
Johannes Pallua, corresponding author
Priv.-Doz. MMag.Dr.rer.nat. Johannes Pallua MSc PhD
Univ.-Klinik für Orthopädie und Traumatologie
Anichstraße 35
A-6020 Innsbruck
Tel.: +43 50 504 80242
Mail: johannes.pallua@tirol-kliniken.at